# Is the Central Nervous System Reservoir a Hurdle for an HIV Cure?

**DOI:** 10.3390/v15122385

**Published:** 2023-12-05

**Authors:** Nazanin Mohammadzadeh, Nicolas Chomont, Jerome Estaquier, Eric A. Cohen, Christopher Power

**Affiliations:** 1Department of Medical Microbiology and Immunology, University of Alberta, Edmonton, AB T6G 2R3, Canada; nazanin3@ualberta.ca; 2Department of Immunopathology, Research Centre of the Centre Hospitalier de l’Université de Montréal (CRCHUM), Montreal, QC H2X 0A9, Canada; nicolas.chomont@umontreal.ca; 3Department of Microbiology, Infectiology and Immunology, Faculty of Medicine, Université de Montréal, Montreal, QC H3C 3J7, Canada; eric.cohen@ircm.qc.ca; 4Department of Microbiology and Immunology, CHU de Québec-Université Laval Research Center, Québec, QC G1V 4G2, Canada; jerome.estaquier@crchudequebec.ulaval.ca; 5Institut de Recherches Cliniques de Montreal, Montreal, QC H2W 1R7, Canada

**Keywords:** HIV, anatomical reservoirs, nervous system, persistence, cure strategies

## Abstract

There is currently no cure for HIV infection although adherence to effective antiretroviral therapy (ART) suppresses replication of the virus in blood, increases CD4^+^ T-cell counts, reverses immunodeficiency, and increases life expectancy. Despite these substantial advances, ART is a lifelong treatment for people with HIV (PWH) and upon cessation or interruption, the virus quickly rebounds in plasma and anatomic sites, including the central nervous system (CNS), resulting in disease progression. With recent advances in quantifying viral burden, detection of genetically intact viral genomes, and isolation of replication-competent virus from brain tissues of PWH receiving ART, it has become apparent that the CNS viral reservoir (largely comprised of macrophage type cells) poses a substantial challenge for HIV cure strategies. Other obstacles impacting the curing of HIV include ageing populations, substance use, comorbidities, limited antiretroviral drug efficacy in CNS cells, and ART-associated neurotoxicity. Herein, we review recent findings, including studies of the proviral integration sites, reservoir decay rates, and new treatment/prevention strategies in the context of the CNS, together with highlighting the next steps for investigations of the CNS as a viral reservoir.

## 1. Introduction

The immunosuppressive lentivirus, human immunodeficiency virus type 1 (HIV), has caused an ongoing global pandemic for the past 40 years. This pandemic has taken over 36 million lives, in large part because patients develop acquired immune deficiency syndrome (AIDS) unless treated with antiretroviral therapy (ART) [1], and upwards of 50% who are not receiving ART may exhibit neurological disorders [2,3]. There is no definitive cure for HIV infection, although ART can suppress replication of the virus, increase CD4^+^ T-cell counts, and reverse immunodeficiency [4]. Despite advances, ART is a lifelong treatment and upon cessation or interruption, the virus rapidly rebounds [5]. Rebound is assumed to originate from specific anatomical and cellular viral reservoirs in which viral genomes persist [6,7]. HIV infects, replicates, and persists in a large number of lymphoid and non-lymphoid tissues, including the brain, gut, lung, liver, heart, kidney, genital tracts, adipose tissue, lymphoid tissue, and bone marrow [8,9,10].

Like other lentiviruses, including simian (SIV), feline (FIV), and bovine (BIV) immunodeficiency viruses, HIV infection of the central nervous system (CNS) occurs shortly after initial systemic infection with detectable virus in the brain, cerebrospinal fluid (CSF), spinal cord, and peripheral nervous system [11,12]. It is widely assumed that HIV enters the CNS by crossing the blood-brain barrier (BBB) or blood-choroid plexus barrier, as a cell-free virus by transcytosis or paracellular entry [13,14,15], or by trafficking into the CNS via infected CD4^+^ T-cells and macrophages (Figure 1) [2,16,17,18]. It is believed that infiltrating CD4^+^ T-cells and perivascular macrophages facilitate infection of proximal microglia, trafficking macrophages, and astrocytes (Figure 1), and HIV-infected CD4^+^ T-cells and macrophages as well as cell-free virus are present in CSF [2,17,18]. Infection is evident by the detection of viral RNA and protein, although the magnitude of viral replication in the CNS is less than that observed in blood or lymphoid tissues [19]. Infection (and activation) of astrocytes and microglia/macrophages, virus-encoded transcripts and proteins, and innate immune induction drives signaling cascades leading to neuroinflammation, synaptic degradation that can culminate in neuronal injury and death (Figure 1) (reviewed in [2]).

## 2. Consequences of HIV Infection in the Brain

HIV infection of the CNS contributes to the development of neurological complications in approximately 25% of people with HIV (PWH) despite available ART [20,21,22]. The spectrum of primary CNS complications ranges from HIV-associated neurocognitive disorder (HAND) to aseptic meningitis, seizures, vacuolar myelopathy, headache, and different movement disorders [2,3]. Of note, opportunistic infections of the nervous system (e.g., Toxoplasmic encephalitis, Cryptococcal meningitis, progressive multifocal leukoencephalitis, primary CNS lymphoma) as well as vacuolar myelopathy and aseptic meningitis are rare among PWH who are receiving contemporary ART. HAND remains an ongoing problem despite available ART and is comprised of three stages of severity: (1) asymptomatic neurocognitive impairment (ANI), (2) mild neurocognitive disorder (MND), and lastly (3) HIV-associated dementia (HAD), which is infrequent in the ART era [2,16]. HAND broadly manifests as memory loss, attention deficit, poor coordination, and slowed psychomotor speed [23], although this classification is increasingly questioned because of the growing appreciation of comorbidities, substance use, and ageing, as cofactors for neurocognitive impairments [3]. The neurological deficits observed in HAND arise due to neuronal damage (synaptic retraction, often followed by cell death) in the cortex, striatum, and limbic system, impacting the lives of PWH and their caregivers by lowering the quality of life, and limiting their daily activities including employment options, and perhaps life expectancy [24,25,26,27,28,29]. The diagnosis of HAND is exigent because of the impact of comorbidities and the overlap with age-related neurocognitive disorders (e.g., Alzheimer’s and Parkinson’s diseases, vascular dementia). Currently, there are no established clinical biomarkers for HAND and the diagnosis rests largely on neuropsychological testing, functional status assessments, and qualitative clinical assessments, which can be biased by education level and cultural context [30,31].

## 3. HIV Infection of Brain-Associated Cell Types

Several studies have investigated the role of different HIV-permissive cell types in the brain and HAND [2]. While not a major cellular reservoir in blood, bone marrow-derived macrophages and yolk sac-derived microglia constitute the chief cellular reservoirs for HIV and related lentiviruses (e.g., SIV, FIV, CAEV) within the CNS [32]. The relative contribution of each of these CNS macrophage cell types to the HIV reservoir and associated pathogenesis remains uncertain in part because of limited tools for delineating their embryological/tissue origins. Indeed, the contributions of other CNS macrophage populations (e.g., choroid plexus- and meninges-derived) to the HIV reservoir and pathogenesis is unclear despite being permissive to HIV. Microglia are yolk sac-derived CNS macrophages that, besides their role as phagocytic immune cells, play a regulatory role in brain development, maintenance, and injury repair [33,34]. Microglial dysfunction can impact development, aging, and fitness of the brain [35]. Perivascular or trafficking macrophages are assumed to be bone marrow-derived, entering the brain from the circulation or perhaps the CSF, and typically exhibit a more immune-activated phenotype, which is often accompanied by greater viral production [36,37]. Moreover, monocytes and macrophages are vehicles for the ongoing transport of HIV into the CNS (Figure 1) and they contribute to viral molecular diversity within the CNS (reviewed in [35]). The macrophage cell types in the brain support productive HIV infection and extensive studies have demonstrated their immediate involvement in HAND pathogenesis, HIV persistence, and viral latency within the CNS [38,39,40,41].

In the era of highly effective ART, there is increasing appreciation of T-cell infiltration of the CNS that might potentially include resting memory T-cells serving as another CNS cellular reservoir for HIV (Figure 1) [19,42]. In addition to T-cells, in recent studies using post-mortem brains of ART-treated PWH, approximately 2–7% of astrocytes showed detectable HIV *gag* mRNA and 0.4–5% contained integrated HIV provirus (Figure 1) [43,44,45].

## 4. Consequences of Microglia/Macrophage Infection by HIV

Microglia/macrophages are key participants in HAND pathogenesis through their production of inflammatory molecules and potential loss of maintenance functions after HIV infection [33,35]. Once these cells become infected, they enable the spread of infection within the CNS and affect the function and survival of proximal neurons and astrocytes by secretion of neurotoxic/neurotrophic and inflammatory molecules (e.g., cytokines, chemokines) (Figure 1) [46,47]. Changes in microglial/macrophage homeostasis during infection and activation drive neuronal damage in HIV-infected brains (Figure 1) [46,47]. Brain imaging, in situ hybridization, RT-PCR, RNA-seq, and immunolabeling studies of brains from PWH, have reported elevated levels of microglia/macrophage activation markers (e.g., MHC Class II, CD68, TNF-α, and IL-1α and β) in persons with HAND compared with different control groups [48,49,50]. Exposure to viral proteins including gp120, gp41, Nef, Vpr, and Tat likely mediates microglial/macrophage activation and contributes to neuronal injury (Figure 1) [51,52,53,54,55,56].

Infected/activated microglia can impact neuronal damage and survival via several mechanisms: (1) secretion of proinflammatory cytokines (TNF-α, IL-1β, IFN-α, IL-6, IL-8) [57,58,59], toxic metabolites (i.e., reactive oxygen species (ROS) and inducible nitric oxide synthase (iNOS) [60,61], and chemokines (CXCL10, CXCL12, CCL2) which contribute to the recruitment of circulating monocytes and lymphocytes into the CNS [50,62]; (2) downregulation of neurotrophic factors (i.e., BDNF), that are essential for neuronal growth and survival [63]; (3) inhibition of autophagy in neurons and accumulation of unwanted proteins and pro-apoptotic state in neurons [64]; and (4) activation of astrocytes resulting in disruption of Ca^2+^, K^+^, and excitotoxic glutamate homeostasis, leading to loss of neuronal maintenance (Figure 1) [65,66].

## 5. HIV Persistence in the CNS despite ART

The impact of current ART regimens on the HIV CNS reservoir and accompanying neurological outcomes are manifold. Since the advent of the combination ART in the late 1990s, plasma (and CSF) viral replication has been increasingly and more effectively suppressed. This was followed by a diminished risk of opportunistic infections, bringing the life expectancy of PWH to similar levels as uninfected persons [67,68,69,70]. However, enduring inflammation might promote the emergence of cardiovascular and metabolic disorders as well as cancer, impacting the quality of life. Moreover, while early antiretroviral drugs were toxic to the peripheral nervous system, other ART drugs (e.g., efavirenz, dolutegravir) appear to exert CNS toxic effects and adverse actions in other organs [71,72,73].

From a neurological perspective, the implementation of ART has diminished the severity of HAND from initial HIV-induced encephalitis with overt neurocognitive disabilities (e.g., dementia) to mild and/or asymptomatic neurocognitive impairments. Nonetheless, the prevalence of HAND has remained seemingly unchanged despite a remarkable reduction in neurological disease severity and perhaps incidence [74,75,76,77]. More recently, several studies have shown HIV and SIV RNA and DNA persist in the brains of both HIV-infected humans and SIV-infected non-human primates (NHPs) despite effective ART [19,39,78,79,80,81,82,83,84]. We have demonstrated that PWH who had received effective ART for a minimum of 5 years and until 12 h prior to post-mortem tissue collection had similar levels of brain total and integrated viral DNA as well as RNA compared with untreated or incompletely virally-suppressed PWH [19,39,78,79,80,81,82,83]. Cochrane and colleagues reported that 11% of integrated HIV proviruses in the brains of viremic PWH are intact in contrast to 15.5% in virally suppressed PWH [83]. Gabuzda et al. further confirmed that the intact HIV provirus accounts for approximately 10% of total proviral genomes in the brain in a cohort of virally suppressed and viremic PWH [82]. More recently, it was shown that the frontal white matter of virally-suppressed PWH contained the highest intact proviral levels, although other regions also contained intact proviruses [84]. These findings raise an important question about the transcriptional status of the intact provirus in the brain and its impact on viral persistence as well as neuropathogenesis.

If the intact provirus can produce a replication-competent virus, it might lead to rebound infection and dissemination of the virus into circulation due to ART interruption and/or the presence of drug-resistance mutations (Figure 2). Moreover, the defective proviruses can be rescued through recombination events during viral replication, which would increase the diversity of the replicating population of the virus [85,86]. Indeed, there is evidence from several animal models, including SIV-infected nonhuman primates and HIV-infected humanized mice, that replication competent virus in the CNS can disseminate into the circulation [43,87].

Several groups have attempted to recover replication-competent virus from post-mortem brain tissues of PWH under suppressive ART. In a recent study, HIV was retrieved from the brains of PWH on ART after rapid autopsy [88]. Tang et al. isolated brain myeloid cells from PWH, cultured them ex-vivo and exposed the cells to various latency-reversing agents (LRAs) to induce viral outgrowth. This group reported an exponential increase of viral RNA and HIV p24 in the supernatants of LRA-exposed brain myeloid cells [88]. The retrieved virus was found to be replication-competent after being evaluated for de novo infection of microglia and PBMCs from HIV[−] donors [88]. Production of replication-competent virions upon stimulation implies two main conclusions: first, although intact proviruses are rare and make up ~10% of integrated viral DNA [82,83], they are inducible upon stimulation and yield detectable infectious virions. Second, the intact provirus is latent as opposed to transcriptionally active since viral production was attainable in only LRA-treated cells and not in the absence of LRA treatment.

These findings might seem at odds with other reports in which HIV p24 and SIV p27 proteins were detected using immunofluorescence microscopy in the post-mortem brains of NHPs and PWH receiving suppressive ART [19], indicating ongoing transcription and translation of viral proteins. However, from studies analyzing the integration sites in PBMCs or PBMC-derived CD4^+^ T-cells from PWH, it is assumed that intact proviruses, compared with defective proviruses, are more frequently located in transcriptionally inactive regions [89,90,91]. This observation might be due to immune selection, eliminating the cells with intact proviruses integrated into more transcriptionally active regions and selecting the deeply latent intact proviruses [89,90,91,92]. Hence, the presence of viral proteins might not be a surrogate indicator of the presence of replication-competent viruses, but merely an indication of transcriptionally active, albeit defective proviruses. However, the integration site(s) in the CNS of ART-suppressed versus viremic PWH remains to be characterized.

Besides the immune response, suppressive ART can influence the dynamics of the HIV reservoirs. Studies of human PBMCs have shown that cells containing intact proviruses are short lived, and selectively deleted with effective ART, likely undergoing apoptosis or another type of regulated cell death [81,93,94,95]. The half-life of intact proviruses is four years for the first seven years; and 18.7 years thereafter [81,93]. The half-life of defective proviruses is much slower, 17 years for the first seven years; and 45 years thereafter [81,93]. In contrast to these findings, in a recent study using both intact proviral DNA assay (IPDA) and Quantitative Viral Outgrowth Assay (QVOA), McMyn and colleagues demonstrated that in resting CD4^+^ T-cells from a cohort of PWH on very long term ART, the reservoir decay occurred for the first seven years on ART, but in the subsequent 13 years on ART there was a slow increase in reservoir size with a doubling time of 23 years [96]. This counterintuitive increase in reservoir size was attributed to the clonal expansion of latently infected resting CD4^+^ T-cells. This dual detection method may have advantages over the IPDA alone method, since IPDA is successful at detection of most but not all defective proviruses and lacks insights into inducibility of intact latent reservoirs [96].

The decay rate of HIV in the brain with and without ART has yet to be determined empirically, although our theoretical studies support the gradual decay of the CNS reservoir, assuming CNS macrophages are the principal cellular reservoir in the brain [97].

## 6. HIV Compartmentalization in the CNS and Drug Resistance Mutations

Sequencing of CNS-derived viruses from untreated individuals has identified a distinct HIV population that is genetically and functionally different from plasma and peripheral tissue’s HIV populations in the same individuals [98,99,100,101,102]. The robust distinction is characterized by the *env* gene V1/V2 and V4/V5 hypervariable regions and the extent of this compartmentalization in plasma compared with CSF correlates with neurological disease [103]. CNS-derived viruses largely use CCR5 for cell entry and yet are both T-cell and macrophage-tropic [42,88,104]. Indeed, the presence of CCR5^+^ macrophage-tropic viruses is associated with HIV persistence during ART and in HAND [99,100,104,105]. In a cross-sectional study with 66 PWH participants, most of who were not on ART, Harrington and colleagues demonstrated that HIV compartmentalization in CSF occurs after the primary asymptomatic phase and mostly during the chronic phase [103].

Compartmentalization of HIV within brain tissue might be due to both external and viral factors, and entry of distinct neurotropic transmitted/founder viruses into the CNS at various stages of infection. HIV displays high levels of genetic variation that is constantly evolving, which enables immunological and pharmacological escapes of the virus. Genetic heterogeneity can contribute to the emergence of drug-resistant variants and/or genetic diversity in the replicating population of the virus. Several factors contribute to genetic diversity in HIV: (1) the virus has remarkably high replication and mutation intrinsic rates. In fact, HIV has a mutation rate of 10^5^ mutations/bp/replication cycle due to the lack of proof-reading ability within the virus’s reverse transcriptase. This is coupled with high replication rates that produce 10^11^ virions/day/patient and 10^8^ infected CD4^+^ T-cells [106,107]; (2) these circumstances can be further compounded if a cell becomes super-infected, as the host genome will simultaneously harbor two different proviruses [108,109]. During virion encapsidation one RNA transcript from each provirus can get packaged into a “heterozygous” virion. Upon infection of a new cell with this virion, the reverse transcriptase, due to its low processivity, will jump back and forth between the two RNA templates (template switching), which can result in the synthesis of a recombinant virus [110]; (3) host determinants including selective pressure by host-encoded restriction factors (e.g., APOBEC3). The APOBEC3s are intrinsic host restriction factors that deaminate cytosine to uracil on the negative DNA strand of HIV during reverse transcription [111]. In the absence of an effective virus-encoded Vif protein, which antagonizes APOBEC3s, the induced mutations can increase diversity among the replicating virus [112]. Investigation of HIV diversity in different regions of the brain has confirmed the presence of drug resistance mutations in *pol* gene, even in some ART-naïve PWH, leaving the patients vulnerable to viral replication [113,114].

Our group recently reported elevated levels of established host restriction factors were expressed in the brain, which were augmented by both HIV and SIV infections and that several restriction factors might contribute to HAND pathogenesis [115]. Using RNA-seq followed by qRT-PCR in the cerebral cortex of uninfected (HIV[−]), HIV-infected without pre-mortem brain disease (HIV[+]), those HIV-infected with neurocognitive disorders (HIV[+]/HAND) and those with neurocognitive disorders and encephalitis (HIV[+]/HIVE), we found robust upregulation of *IFITM1* and *MAN1B1* mRNA expression levels in both HAND groups. Machine learning algorithms identified *MAN1B1* as an important variable that distinguished the HAND groups from the HIV[+] group [115].

## 7. CSF Viral Escape Dynamics

While HIV presence in CNS tissue reservoirs and its dissemination to the circulation is difficult to measure, CSF viral escape is a recognized clinical phenomenon and occurs when there is detectable HIV RNA in the CSF in the absence of detectable plasma viral RNA. The discordance between plasma and CSF viral load is attributed to the compartmentalization of HIV in the CNS with associated presumed HIV egress from the CNS tissues into the CSF. The incidence of CSF escape has been reported in between 4–17% of adults receiving ART [116,117,118,119]. Pérez-Valero and colleagues assessed 1264 PWH who were on stable ART for more than six months with plasma viral loads <50 copies/mL. The plasma and CSF viral RNA levels were measured between 2003 and 2011 and HIV CSF escape was reported in 4.4% of the participants [117]. In a recent study with 114 PWH receiving suppressive ART that had CSF examination between 2017 and 2022, the viral escape was estimated at 17% [118]. Besides HIV, CSF viral load for EBV, VZV, CMV, HHV-6, and JC virus was quantified, revealing EBV-encoded nucleic acid detection in 9% of the cohort, which was ascribed to HIV infection-induced CSF pleocytosis although it was unclear whether EBV and HIV CSF escape were temporally concurrent [118]. Filippidis and colleagues assessed 288 PWH who underwent lumbar puncture between 2011 and 2019 and detected HIV escape in 6.5% of the participants who had suppressed plasma viral RNA levels [119].

While there is no consensus on triggering events or the consequences of CSF viral escape, it has been associated with protease inhibitors use, although the low frequency of this phenomenon among ART-receiving PWH suggests the involvement of secondary triggering factors [117,120,121]. Low-level persistence of plasma viremia during ART is another potential contributor to the presence of drug-resistant viral variants in the CSF [122,123,124]. The impact of CSF escape on neurocognitive performance remains uncertain; some groups have associated the presence of HIV in the CSF with neurocognitive decline [49,120,125,126], although these associations were not confirmed by other studies [117,118,119]. Another unknown aspect of HIV CSF escape is whether the escaped virus can contribute to systemic rebound infection, or merely prompt transient plasma viral elevations (termed ‘blips’) that can resolve with adherence to the prescribed ART regimen.

## 8. Impact of ART on HIV Persistence in the Brain

The brain’s unique and complex anatomic structure makes this organ a major hurdle for the HIV cure endeavor. The BBB mediates the penetration of ART into the brain [23,127,128]. The lower efficacy of select ART drugs in limiting the replication of HIV in microglia/macrophage, compared with blood-derived lymphocytes, can promote virus seeding and persistence within the brain (Figure 2) [19,129]. There are currently no specific ART regimens targeting individual anatomic viral reservoirs. Nonetheless ART modification and/or intensification might ameliorate HAND severity, thereby increasing the quality of life for PWH (e.g., maraviroc) [130].

Historically, a CNS penetration effectiveness score (CPE) was assigned to each antiretroviral drug [131]. This scoring system was based on each drug’s molecular weight, CSF concentrations, *in vitro* IC50 in non-neural cells, and clinical reports of efficacy in controlling CSF viral RNA levels [131]. This scoring system was revised based on stronger association with CSF viral load, with higher CPE scores suggesting higher CNS penetrance and lower CSF viral load [132]. This scoring system served as a practical guide for regimen alteration in response to neurological status, although this scoring system depends on ART drug concentrations in CSF and does not capture neural cell type-specific efficacies of ART drugs. Recent studies could not draw a positive or negative association between CPE scores and neurocognitive impairments [133,134,135].

Studies of ART drugs levels in brains from animals and humans report a broad range of concentrations and differing levels of effectiveness that were contingent on the cell type and individual ART drug [19,129]. Based on the findings that replication-competent virus isolated from postmortem brains was CCR5 tropic and from a pure microglia culture, use of a CCR5 antagonist drug, such as maraviroc with higher efficacy in monocytes, may be suitable for regimen change/intensification if neurocognitive impairment emerges [130,136,137]. In a recent study, Letendre and colleagues showed that in a 96-week trial of ART intensification with dolutegravir and maraviroc, dolutegravir and placebo, or dual placebo, empiric ART intensification was not effective as a treatment for neurocognitive impairment [138].

Long-acting ART drugs, including cabotegravir, an integrase strand transfer inhibitor, and rilpivirine, a non-nucleoside reverse transcriptase inhibitor, are prescribed as two intramuscular injections in-clinic every other month or monthly and represent an alternative to a daily oral ART regimen [139,140]. Although this regimen allows for more stringent adherence, both as a pre-exposure prophylaxis (PrEP) and treatment, the impact on brain viral load and CSF load or escape is unknown and yet to be investigated.

Currently, small-molecule ART drugs are the only approved treatment for HIV infection. There are several clinical trials of broadly neutralizing Abs (bnAbs) in PWH (reviewed in [141,142,143,144]). These potent Abs can neutralize a wide range of HIV variants by interacting with conserved regions of the HIV envelope glycoprotein trimer and are promising for use in the prevention and treatment of HIV [141,143]. The use of monoclonal antibodies for the treatment of CNS cancers has been shown ineffective because the concentration of these antibodies in the brain and CSF is lower than of that in serum [145]. Moreover, the unique immune privileged status of the CNS might inhibit bnAbs from exerting their full inhibitory effects via antibody-dependent cellular cytotoxicity (ADCC). The efficacy of bnAbs immunization in prevention and elimination of CNS HIV infection is unknown, which might influence their use in HIV cure initiatives.

Inspired by the cancer immunotherapy field, chimeric antigen receptor (CAR) T-cells are another promising HIV therapy [146,147]. In this therapeutic approach, PWH’s T-cells are isolated and genetically engineered to express CARs via a lentiviral delivery vector system. This intervention enables CAR T-cells to recognize and lyse HIV-infected cells, mediating their eventual clearance [148,149,150]. The technical challenges of this approach include uncertainty regarding the consequences of lentiviral delivery to potentially latently infected T-cells from PWH, detection of latently infected cells with a translationally silent virus, and unknown status of access to deep tissue reservoirs (reviewed in [151]).

Gene therapy based on transplantation has been a successful HIV cure strategy, in which a donor’s CCR5-delta-32 mutation autologous stem cells are infused into an immunoablated PWH; this form of transplantation has been performed successfully in a very limited number of PWH who have had hematologic cancers, requiring allogeneic stem cell transplantation, and they are seemingly virus-free after this procedure [152,153].

## 9. Discussion

The contribution of the CNS reservoir to HIV rebound infection can be appreciated from case reports of two PWH who received allogeneic hematopoietic stem cell transplantation without the CCR5-delta-32 mutation and remained HIV negative for months after ART cessation [154]. These individuals experienced HIV rebound at 12 and 32 weeks after ART cessation, evidenced by plasma viremia [154]. In fact, HIV was detected in the CSF from both patients, and one patient experienced CNS symptoms before detectable HIV rebound in the plasma [154]. Moreover, the most recent attempt to retrieve replication-competent virus from post-mortem brains of PWH receiving ART with undetectable plasma viral load yielded successful isolation of the virus from microglia [88]. This replication-competent virus was retrieved after histone deacetylase inhibitor (HDACi) exposure. The derived virus was then used to infect CD8^+^ T cell-depleted PBMCs from healthy (uninfected) donors [88]. The absence of detectable virus without LRA treatment underscores the notion that the HIV brain reservoir is latent and in the absence of LRA stimuli, which is an unnatural circumstance in the brain, this reservoir remains dormant. Thus, it is crucial to determine if the brain viral reservoir is capable of autonomous activation of viral replication by cytokines and/or stimulatory molecules present in the brain, that could prompt translocation of the virus to the circulation via infection of trafficking PBMCs.

To move toward a functional HIV cure, it is imperative to reduce the size of all viral reservoirs throughout the body. Given the current understanding of HIV infection of the CNS, a viral reservoir within the CNS is a major barrier to HIV eradication. To reduce the viral pool in brain tissue, a deeper understanding of the determinants that drive HIV latency in HIV-infected cells in the brain is essential. Further investigation is pivotal for a fuller grasp of the viral integration sites in different anatomic sites and cell types of the brain, the impact of the duration of therapy on these sites, and the corresponding viral decay rates in both cellular and anatomic CNS reservoirs.

An equally challenging issue is the genetic diversity and compartmentalization of CNS macrophage-derived viruses versus the entire CNS-associated viral population (including viruses derived from astrocytes and infiltrating T-cells). Moreover, the replication and tropism properties of HIV found within different regions of the brain have yet to be investigated fully.

Persistence, latency, low ART penetrance of the CNS, lack of a local adaptive immune response and limited efficacy of ART in inhibiting viral replication in microglia/macrophage make the CNS an impregnable reservoir for cure. Stimuli like LRAs, intercurrent infections, drug resistant variants, and interrupted treatment may potentially induce the replication of intact replication-competent latent virus that can egress from the brain to the circulation (Figure 2). With the burgeoning efficacy of ART and the shift of focus toward increasing the quality of life for PWH, it is of urgent importance to involve, consider, and reconsider different anatomical and cellular reservoirs in HIV cure, treatment, and prevention research.

The next call for action is for the inclusion of underserved PWH populations (e.g., children, substance users, as well as those PWH who are socially or economically marginalized) in clinical and basic science research. Recent studies of adolescents living with HIV have demonstrated that this group suffers from accelerated epigenetic aging with their mean DNA methylation age (16.01 years) being higher than their mean chronological age (10.77 years) [155]. This accelerated biological aging is associated with poorer cognitive functioning and modulation of brain volume and cortical thickness in different brain regions [155,156,157].

Accelerated brain aging has also been observed with substance (e.g., opioid) use and HIV syndemic (reviewed in [158,159]. Besides the exacerbating impact of opioids on neuroinflammation, neuropathology, and immune modulation, opioids can worsen HIV infection through immunosuppression and facilitate viral seeding of the CNS by damaging the BBB integrity, increasing recruitment and activation of myeloid cells, via the release of chemo-attractants from astrocytes, enhancing CCR5 and CXCR4 expression (reviewed in [160]). With an ongoing opioid crisis and the synergic effects of HIV and opioids syndemic, it is key to establish a cohort of PWH with substance use histories for future HIV-persistence investigations.

In summary, despite the remarkable advances in HIV care and outcomes with ART over the past four decades, multiple hurdles stand in the way of a cure for HIV. These obstacles include the preponderance of HIV-infected macrophage type cells and limited ART efficacy in CNS as well as a full understanding of HIV neurobiology (Figure 2). With over 37 million PWH globally, many living in very trying socio-economic circumstances and often without access to ART, there remains an urgent and immense need to address these gaps in the knowledge and care of HIV infection of the CNS.

## Figures and Tables

**Figure 1 viruses-15-02385-f001:**
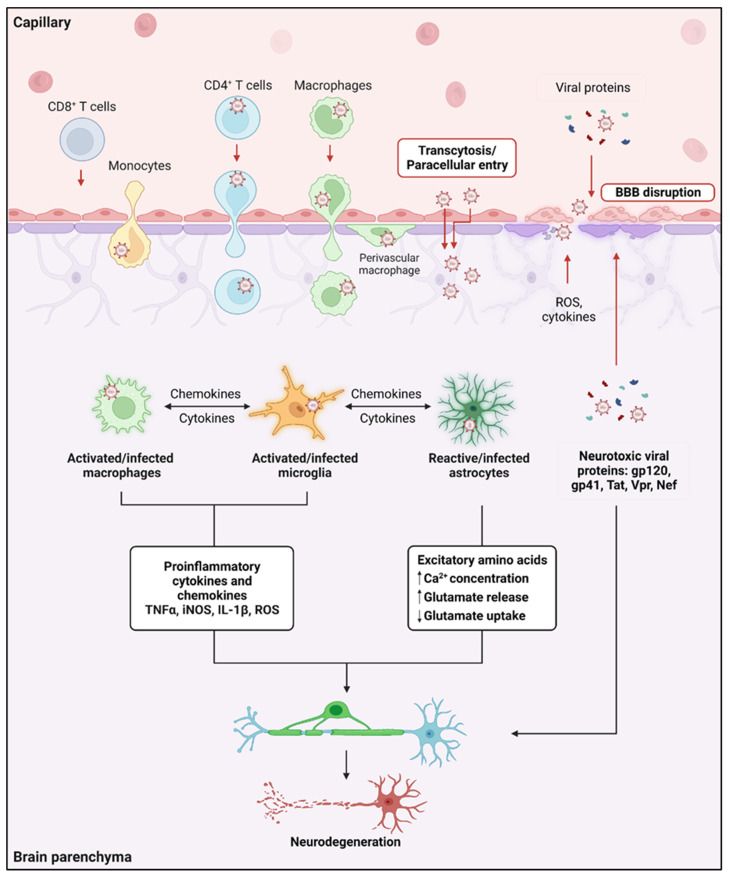
HIV neuropathogenesis. HIV-1 enters the brain as a cell-free virion or via HIV-infected myeloid cells and CD4^+^ T-cells that cross the blood brain barrier (BBB). In the brain parenchymal environment, HIV-1 infects perivascular macrophages, microglia, and astrocytes. Infected cells release inflammatory cytokines, chemokines, and neurotoxic viral proteins that can influence the permeability of the BBB, recruit immune cells to the site of infection, and disrupt neurotransmitter (e.g., glutamate) and Ca^2+^ homeostasis that results in neuronal injury and eventual death.

**Figure 2 viruses-15-02385-f002:**
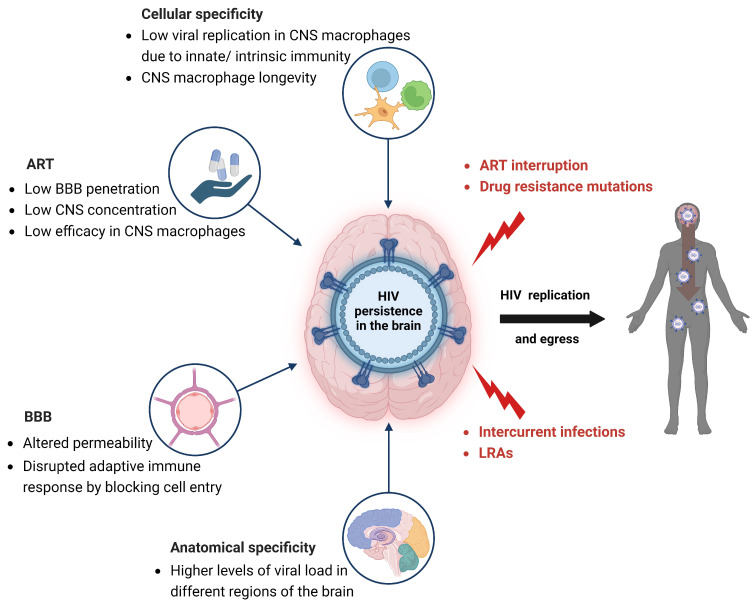
Dynamics of HIV infection and persistence in the brain. HIV-1 DNA, RNA, and proteins persist in the brain despite effective antiretroviral therapy (ART) and concomitant immune responses. Multiple factors contribute to HIV persistence in the brain despite ART including, lower levels of HIV replication in central nervous system (CNS) macrophages, as well as CNS macrophage resistance to HIV-induced cell death and associated long half-lives. Reduced ART efficacy in these cells is amplified by low ART drug concentrations in the CNS due to limited blood-brain barrier (BBB) penetration. The BBB also blocks entry of B and T-cells leading to the comparative lack of adaptive immune responses in the CNS. The direct and indirect effects of HIV infection can alter the BBB’s permeability, enhancing the trafficking of infected cells. Lastly, higher levels of HIV burden in select neuroanatomic sites might contribute to increased seeding or perhaps diminished clearance of the virus in these sites, which results in persistent viral reservoirs in the brain. Upon stimulation of the brain’s latent HIV reservoir by ART interruption, latency reversing agents (LRAs), drug-resistant viral variants, or intercurrent infections, intact provirus may produce replication-competent virions that can potentially egress into the circulation.

## Data Availability

Not applicable.

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
