# Peer review of "Is the Central Nervous System Reservoir a Hurdle for an HIV Cure?"

_viruses, 2023, doi:10.3390/v15122385_

Round 1
Reviewer 1 Report
Comments and Suggestions for Authors
VIRUSES
RE: Is the Central Nervous System Reservoir a Hurdle for an HIV Cure?
I appreciate the opportunity to review this well written and interestingly dense article on the brain/CNS being a reservoir for HIV. I am a cognitive researcher in HIV and I read this will a great deal of interest as my own work intersects closely with this. Overall, I found this article very detailed, and although the information was very dense and intricate, I was able to follow. For my work, this is a great reference article for me. Unfortunately, I don’t have anything to add in terms of making it better. The deep virology is beyond my own area of expertise, but as someone in the area, I can comment that this is a very useful and informative article, one that I would like to read a few times to soak in the knowledged.
Author Response
We are grateful for your attention to detail and are pleased that you found this review useful as a cognitive HIV researcher. We have tried to include the most updated information on different aspects of HIV infection and persistence in the CNS including clinical and basic science data.
Reviewer 2 Report
Comments and Suggestions for Authors
This is a short, well-written review of some of the issues in the understanding of the brain as a potential reservoir for HIV in individuals who are on ART and particularly those who are virologically suppressed. The reference list is comprehensive without being overwhelming. While I would probably emphasize some sections more prominently and not others, these are individual authorial decisions. Sometimes the other issues (therapy, etc.) are discussed in more detail than the important papers that relate more directly to the title. Nevertheless, I have a few comments for their consideration:
1. There is a good summary of the current understanding of neuropathogenesis. The issue of viral protein toxicity, particularly proteins that are not present extracellularly in high (or even moderate) quantities persists in the literature. In view of the contemporaneous emphasis of this review, the authors should not perpetuate these ideas since they are not considered viable by mainstream pathogenesis investigators.
2. Similarly the prevalence of HAND in virally suppressed individuals is a controversial subject that depends on whether one considers ANI as HAND. Since the definition of ANI is also difficult (one or more domains, etc.), I personally do not believe that the prevalence of HAND has not dropped with ART. It is significant to those who have it, of course, but it is probably lower.
3. Similarly I would not spend a lot of time on CPE scores. This has been difficult to replicate.
4. CCR5 inhibitors have been used clinically. It might be more useful to discuss these studies than the CPE scores if CNS specific therapy is a part of the review.
Author Response
- There is a good summary of the current understanding of neuropathogenesis. The issue of viral protein toxicity, particularly proteins that are not present extracellularly in high (or even moderate) quantities persists in the literature. In view of the contemporaneous emphasis of this review, the authors should not perpetuate these ideas since they are not considered viable by mainstream pathogenesis investigators.
In light of recent studies highlighting that the majority of persistent proviruses are defective and yield non-replication competent viruses during ART (Cochrane et al., Ann Neurol 2022, Gabuzda et al., Viruses 2023, and Mohammadzadeh et al., mBio 2021) yet viral proteins have been detected in the brain, it is plausible that viral proteins from defective viral genomes are encoded and exert neurotoxic effects. Given that this is a review paper and not an opinion article, it seems most fitting to offer the readers all considered mechanisms.
- Similarly the prevalence of HAND in virally suppressed individuals is a controversial subject that depends on whether one considers ANI as HAND. Since the definition of ANI is also difficult (one or more domains, etc.), I personally do not believe that the prevalence of HAND has not dropped with ART. It is significant to those who have it, of course, but it is probably lower.
We agree with the reviewer regarding the prevalence of HAND during ART but unfortunately, the data supporting this contention are missing, and thus, more work needs to be done on the post-ART definition, incidence, and prevalence of HAND.
- Similarly I would not spend a lot of time on CPE scores. This has been difficult to replicate.
We have discussed the CPE scoring system to give the readers a historical perspective on ART and in some quarters, CPE continues to be discussed. We have discussed the CPE scoring system in this paragraph (Page 13, lines 20-23 and page 14, lines 1-5) concluding that this system does not capture neural cell type-specific efficacies and recent studies could not replicate the historic data.
- CCR5 inhibitors have been used clinically. It might be more useful to discuss these studies than the CPE scores if CNS specific therapy is a part of the review.
We have addressed this issue (page 14, lines 8-11). Of interest, a recent study by Letendre., et al Clinical Infect Dis 2023, in a 96-week trial of ART intensification with DTG + MVC, DTG + Placebo, reports that empiric ART intensification is not effective as a treatment for neurocognitive impairment.
Reviewer 3 Report
Comments and Suggestions for Authors
This is a review that analyzes the role of the nervous system as a reservoir for HIV and its possible limitation for the cure of the infection. The review is well structured but the contribution to the existing literature is limited. Throughout the text no possible hypotheses, alternatives, interventions, etc. are proposed. In the text there are paragraphs with different style. In the bibliography there are citations that are not adequately referenced (i.e., 38, 95...).
Comments on the Quality of English LanguageNone
Author Response
Given that this is a review paper, we have addressed the contemporary issues and data, germane to HIV infection of the CNS and its treatment. A review of speculative hypotheses or alternative therapies is not warranted in this type of article. We have corrected the references as per the reviewer’s request.
Round 2
Reviewer 3 Report
Comments and Suggestions for Authors
None
Comments on the Quality of English LanguageNone